# Revolution in Cancer Treatment: How Are Intelligently Designed Nanostructures Changing the Game?

**DOI:** 10.3390/ijms25105171

**Published:** 2024-05-09

**Authors:** Désirée Gül, Burcu Önal Acet, Qiang Lu, Roland H. Stauber, Mehmet Odabaşı, Ömür Acet

**Affiliations:** 1Department of Otorhinolaryngology Head and Neck Surgery, Molecular and Cellular Oncology, University Medical Center, 55131 Mainz, Germany; burcuonal@asu.edu.tr (B.Ö.A.); qianglu@uni-mainz.de (Q.L.); rstauber@uni-mainz.de (R.H.S.); 2Chemistry Department, Faculty of Arts and Science, Aksaray University, Aksaray 68100, Turkey; modabasi@aksaray.edu.tr; 3Pharmacy Services Program, Vocational School of Health Science, Tarsus University, Tarsus 33100, Turkey

**Keywords:** nanoparticle design, cancer, treatment, nanotheranostics, drug delivery

## Abstract

Nanoparticles (NPs) are extremely important tools to overcome the limitations imposed by therapeutic agents and effectively overcome biological barriers. Smart designed/tuned nanostructures can be extremely effective for cancer treatment. The selection and design of nanostructures and the adjustment of size and surface properties are extremely important, especially for some precision treatments and drug delivery (DD). By designing specific methods, an important era can be opened in the biomedical field for personalized and precise treatment. Here, we focus on advances in the selection and design of nanostructures, as well as on how the structure and shape, size, charge, and surface properties of nanostructures in biological fluids (BFs) can be affected. We discussed the applications of specialized nanostructures in the therapy of head and neck cancer (HNC), which is a difficult and aggressive type of cancer to treat, to give an impetus for novel treatment approaches in this field. We also comprehensively touched on the shortcomings, current trends, and future perspectives when using nanostructures in the treatment of cancer.

## 1. Introduction

Engineered nanomaterials (NMs) suggest a big potential in enhancing the accuracy of disease diagnosis and treatment. By utilizing nanotechnology, the drawbacks of conventional delivery methods can be overcome. Some of the limitations encompass difficulties at a broader level, like biodistribution, along with obstacles at a more specific level, such as intracellular trafficking. Nanotechnology facilitates the targeting of cells, the transportation of molecules to specific organelles, and various other groundbreaking endeavors. In order to accelerate the implementation and practical use of emerging nano-enabled technologies, the National Nanotechnology Initiative (NNI) was launched by the US National Science and Technology Council (NSTC) in the year 2000. This initiative outlined well-defined strategies and ambitious goals for the field. The recent endeavors aimed at exploring and enhancing nanotechnology have been bolstered by these programs, with nanoparticles (NPs) representing a substantial segment of the research and progress documented [1].

NPs possess the capacity to enhance both stability and solubility of enclosed cargos, make their transportation across membranes easy, and extend their circulation durations, thereby boosting both safety and effectiveness [2,3]. NP research has been extensive due to these factors, producing encouraging outcomes in laboratory settings and in small animal studies [4]. Nevertheless, in spite of the comprehensive research conducted in response to the NNI, the actual number of nanomedicines accessible to patients falls significantly short of the anticipated figures for this domain. This discrepancy can be attributed, at least in part, to a translational disparity that exists between studies conducted on animals and those conducted on humans [4,5]. More specifically, it is crucial to figure out how these variances impact the behavior and functionality of nanomedicines within the human body [6]. The limitations in clinical translation are not solely attributed to the variations observed among different species. The presence of heterogeneity among patients can also impede the efficacy of nanomedicines. Furthermore, the research conducted on the interactions between nanomedicines and specific patient populations is currently limited. As a result, only a limited number of nanomedicines that have been approved are proposed as first-line therapy choices, and their effectiveness is observed in only a small subset of patients [7]. The lack of comprehensive research on the diversity present in both the biological mechanisms of diseases and in patient populations contributes significantly to this phenomenon. This variability impacts the efficacy of NPs due to changes in the growth, composition, and function of diseased tissues affecting the distribution and performance of NPs [1].

The escalating prevalence and fatality rate of patients with cancer is a primary contributor to global mortality. Each year, approximately 14 million fresh instances and nearly 8 million fatalities are documented due to this disease [8]. Various traditional methods such as surgery, chemotherapy, radiotherapy, and targeted therapies are commonly utilized in the treatments of patients with cancer, often resulting in suboptimal outcomes [9,10,11]. Due to constraints in conventional treatment methods, researchers are exploring novel strategies for cancer treatment that possess superior targeting capabilities, favorable results, and minimized side effects [12]. In order to boost the efficacy of tumor treatment, it is essential to develop a drug delivery system (DDS) that may specifically target tumor tissues, thereby maximizing the therapeutic effects of the drug while minimizing damage to healthy cells and organs. The precise and organized delivery of NPs that specifically target cancer cells is able to overcome a range of biological and physiological obstacles. This ultimately leads to the direct penetration of the tumor microenvironment. The barriers possess an intricate composition comprising numerous layers and components of a physiochemical and enzymatic nature. Specific targeting is also required by factors such as the sizes of NPs, their surface charges, and their surface chemistries. Furthermore, the development of theranostic NPs has allowed for the integration of diagnosis and treatment within a unified system (Figure 1). This innovative approach enables a more individualized treatment strategy by delivering precise doses, as well as allowing us to monitor the distribution, targeting, and response to therapies through the use of an imaging tool [13].

Nanoparticles possess the ability to circumvent drug-resistant mechanisms present in cancer cells, thereby facilitating the delivery of drugs to their desired destinations and exerting therapeutic impacts. By manipulating NPs, it is possible to regulate the release of drugs in a controlled manner, ensuring consistent drug concentrations and reducing adverse reactions. Additionally, current research endeavors are concentrated on enhancing the precision of NP-based systems, enabling them to selectively target cancer cells while preserving healthy tissues. This advancement holds immense potential for the advancement of personalized and efficient cancer therapies in the forthcoming years [14].

Certain cancer cells are able to exhibit distinct qualities that render them less susceptible to NP therapies. Furthermore, prior to the widespread implementation of NP-based treatments in clinical settings, it is imperative to thoroughly evaluate the potential for off-target effects and unintended outcomes. Additionally, the manner in which NPs are delivered to cancer cells plays a pivotal role and should be carefully considered. The efficacy of these NPs heavily relies on their ability to efficiently reach and penetrate the tumor site [15]. Furthermore, it is imperative to grasp the potential long-term consequences and safety characteristics of NP-derived therapies to ensure the welfare of patients and minimize any unanticipated hazards [16]. The interplay between NPs and immune cells, as well as the efficient delivery of therapeutic agents, is significantly affected by the sizes, shapes, and surface characteristics of these NPs. This underscores the pivotal role played by NPs in the efficacy of immunotherapies [17]. Science is working towards having the ability to improve the structure of NPs to increase the efficiency of targeting and the overall effectiveness of immunotherapy. It is crucial to consider stability and biocompatibility to reduce toxic effects and guarantee the long-term safety of patients. Biodegradable polymers, for instance, provide a secure and regulated release of therapeutic substances, enabling precise delivery to particular immune cells or tumor locations [18].

Some minor modifications, such as an adjustment of surface chemistry, nanoparticle shape, NP size, and NP-protein corona (PC) form, and the use of non-toxic and biodegradable structures, can be extremely rational and effective in cancer treatment. For this purpose, in this review, we touched upon the important points to consider in NP design, the importance of NP morphology, some important and special nanostructures, applications in cancer treatment, and future expectations. Moreover, HNCs encompass a diverse group of aggressive malignancies that are difficult to treat due to their genetic complexity. Therefore, in this review, we included the applications of this difficult and common type of cancer as a model in recent years.

## 2. Selection and Design Nanostructures

Recent progress in cancer diagnosis and treatment can be linked to the physicochemical properties of NPs and the capacity to tailor the surface of NPs for specific uses [19,20]. NPs are defined as small particles ranging from 10 to 500 nm in size, which are synthesized using chemical-physical methods and are composed of diverse materials such as organic, inorganic, and biological substances [21]. Until now, the FDA has granted approval for a limited number of NMs to undergo clinical trials. The most significant of these include liposomes, albumin-based nanostructures, iron-based NPs, PLGA-based NPs, and silica NPs [7,22]. In addition, the utilization of NPs in medical contexts necessitates the evaluation of distinct factors, including the chemical composition, average size, shape, and morphology of the particles, as well as the distribution of particle sizes and their both chemical and physical stabilities. The utilization of NPs in clinical settings has been authorized by the FDA through diverse approaches. These include oral administration for imaging objectives, local administration for the transportation of peptides and other small molecules, topical application to surmount skin barriers, and systemic administration for the treatment of different types of cancers and other ailments [23].

Substantial endeavors persist in the pursuit of creating and advancing polymeric nanocarriers that possess customized physical, chemical, and biological characteristics. The sizes, shapes, and surface characteristics of NP act a critical role in influencing their circulation in the bloodstream, distribution within the body, cellular uptake mechanisms, intracellular positioning, and overall accessibility for biological interactions [24]. A highly effective approach to prevent NPs from interacting with BF components involves applying a thick layer of neutral hydrophilic flexible polymers as a coating. This layer serves to inhibit the adsorption of plasma proteins, thus preventing recognition by the mononuclear phagocytic system (MPS) and subsequent clearance. In order to enhance the longevity of stealth NPs, the conventional approach involves applying a layer of nonionic surfactants like poly(ethylene glycol) (PEG) onto them. The ethylene glycol units create a hydration layer by closely binding with water molecules, thereby inhibiting protein adsorption and subsequent clearance. Consequently, this mechanism effectively extends the lifespan of the particles within the circulation system [2]. Nevertheless, PEG does not possess the capability to entirely inhibit protein adsorption onto NPs. Consequently, it becomes imperative to fine-tune the density and length of PEG molecules in order to attain the most favorable antifouling outcome. In a similar vein, zwitterionic coatings consist of molecules that possess balanced charges, thereby establishing a surface that is neutral in nature. These coatings employ well-organized layers to construct a water-based barrier that efficiently hinders protein interference [25]. However, it is crucial to take into account the foreign characteristics of the artificial antifouling polymers. For example, there have been reports of an acquired immune response to the PEG moiety, which can potentially hinder the performance of PEG-NPs [26].

The acknowledgment of the wide-ranging influence of the surface of a nanosystem, encompassing its outermost layer, on its physical, chemical, and biological features is well established. Moreover, coatings have a substantial influence on the interaction between the nanosystem and its surroundings, ultimately impacting the delivery process of NPs in vivo. A thorough understanding of the characteristics and coating attributes, as well as their influence on the behavior of NPs, is crucial for the progress of reliable and effective nanotechnologies [2,27].

The enhancement of biopharmaceutical agents’ quality stands as a paramount principle within the “Quality by Design” framework. This principle encompasses not only the anticipated therapeutic effects of the novel product but also its pharmacokinetics, pharmacodynamics, and potential toxicity [28]. The existing requirements for cancer treatment could be fulfilled through the enhancement of contemporary techniques for drug administration [29]. In recent times, despite extensive exploration for cancer treatment, researchers have been actively seeking diverse DD techniques encompassing a range of forms [30].

Polymeric nanoplatforms have garnered significant interest, primarily owing to their versatile core–shell structures that can be effectively employed for both physical and chemical processes [31,32,33,34]. Enhancement of drug solubility, suitable impact on drug pharmacokinetics, and enhancement of permeability and retention (EPR) properties are key considerations in the formulation of DD techniques. The utilization of polymeric nanostructures that can react to external stimuli is regarded as highly promising for the advancement of DDSs [35]. Recently, there has been a notable rise in enthusiasm for smart/stimuli-responsive nanostructures [36,37,38]. Most tumor cells possess a distinct characteristic that enables the selective aggregation of polymeric nanocarriers with specific sizes within tumor cells, unlike normal tissues [34]. In this case, synthesis of appropriate NMs according to the need/situation is also mandatory.

Besides dimensions and surface properties, the morphology of NPs has emerged as a crucial element impacting factors such as circulation duration, biodistribution, cellular absorption, and specificity in the context of delivering cancer drugs. The majority of nanocarriers created for delivering anticancer medications are typically manufactured in a spherical shape. Conversely, viruses and bacteria can be found in a range of shapes, including filaments or cylinders. It is noteworthy that viruses and bacteria have adapted to exist in nonspherical forms, enhancing their capacity to avoid detection by the immune system. It is increasingly acknowledged that the field of nanomedicine should draw inspiration from natural biological systems, recognizing that NPs molded into nonspherical configurations could potentially offer unique advantages compared to nanospheres of similar size [39].

The small dimensions of NPs, along with their surface chemistry, are recognized for providing advantageous pharmaceutical characteristics, and they could also play a role in causing harmful effects. Reduced clearance of NPs can result in enhanced cellular penetration for smaller NPs in comparison to larger molecules. However, this may also lead to the retention of some particles in the body. In the event that a highly active or cytotoxic NP is not eliminated during the initial pass through the body, but is rather retained, there is a possibility of adverse effects occurring at the intended site due to unintended retention. Cytotoxic drugs, when administered systemically, have the capability to exert their cytotoxic effects on tissues during the first pass, before reaching the intended target tissues. Globally, around 70% of manufactured medications exhibit low solubility in water, leading to less-than-ideal pharmacokinetic properties within a living organism. NP DDSs were created in order to tackle this problem by enabling medicinal compounds to be transported precisely and effectively, thereby preventing damage to surrounding organs that may occur if the drugs were administered in their free form [40].

An Important group of NMs is prot”in-based nanostructures. Given that various biological media or organisms are implicated, it is imperative for the engineered NMs to possess favorable biosafety and biocompatibility. Consequently, the choice of nanomaterial precursors or components becomes essential [41]. Proteins are viable options due to their essential role in the functioning of living organisms [42]. Proteins possess the ability to interact with both hydrophobic and hydrophilic components due to their amphiphilic nature. These proteins can be referred to as “participants” as they fulfill diverse functions in nanomaterials. They bestow a multitude of inherent characteristics upon protein-based NMs and nanosystems [43]. Protein nanocomposites, which encompass protein NPs and their conjugates, have emerged as sophisticated nanocarrier systems [44]. Protein NP synthesis is a cost-effective process that employs straightforward techniques, which can be adapted to achieve the desired size distribution [45].

Protein NPs, being both non-toxic and smaller in size compared to conventional NPs derived from metals or other inorganic materials, are efficiently eliminated from the human body [46]. Studies have demonstrated their ability to elicit minimal immune reactions in humans [47]. Moreover, certain alterations confer exceptional stability to these NPs throughout storage and in vivo procedures [48]. Protein NPs offer the advantage of being modified with functional groups that aid in the targeted delivery of drugs and other beneficial compounds to precise sites [49]. Protein NP conjugates have been effectively employed by researchers in various applications, apart from protein NPs [50].

It is becoming increasingly important to develop effective strategies to manipulate the shape, size, and surface properties of nanostructures. An important point to remember is that progress in cancer research and academic research institutions is still in its infancy. Therefore, it is imperative to adopt a more rational and long-term approach to overcome the challenges in cancer design and nano research. In future cancer research, it would be very useful to focus on the heterogeneity of human tumors and, in particular, the design of tailored nanoplatforms for personalized cancer treatments [51].

## 3. The Protein Corona—A Crucial Factor in Nanomedicine

Nanoparticle drug delivery systems have shown promise in delivering therapeutics effectively within natural environments, including the complex milieu of the human body. In this natural environment, NPs encounter various physiological challenges, such as interactions with blood components, immune responses, and tissue-specific barriers. Their interactions with biomolecules are significantly influencing NP-based applications because they can change their biological identity.

Due to their high surface free energy, nanostructures interact with proteins upon introduction into biological fluids (BFs). These interactions, including electrostatic, Van der Waals, hydrophobic, and hydrogen bonding, drive the adsorption of proteins onto the surface of NPs in the biological environment [52]. These protein layers were defined as the “protein corona (PC)” [53]. The PC can be categorized into different layers, encompassing the ‘soft’ and ‘hard’ corona, based on the binding affinity between the proteins and the NPs’ surface [54]. The formation of PCs in biological fluids is a very dynamic process. Proteins with low affinity and reversible binding, such as serum albumin [55], may be replaced by proteins with low abundance but high affinity, such as immunoglobulins [56], apolipoprotein ApoA [57], and complement factors [58]. The phenomenon known as the “Vroman effect” involves a time- and concentration-dependent exchange to achieve dynamic equilibrium. The NP-protein complex eventually achieves a stable and irreversible minimum energy state as time progresses [55,59,60].

Throughout the adsorption process, proteins have the potential to experience conformational changes as they adjust to the surface of the nano-biosystem. This event possesses the capacity to induce modifications in the secondary/tertiary structures of proteins and their interactions with the surrounding media. The biological functions of proteins in BFs rely on their interactions with various biomolecules and substances. Therefore, even minor alterations in protein structure resulting from interactions with NPs can greatly affect their pharmacological activities and interactions with other proteins, resulting in varied biological responses [59].

The existence of a responsive PC can significantly influence the uptake of NPs by cells and the activation of intracellular pathways [61]. PC modification affects various physical and chemical attributes of NPs, including their surface charge, dimensions, and state of aggregation. As a result, these modifications influence the biological fate of NPs, affecting their distribution within the body, how they are processed in the body, and their therapeutic impacts. It has been well known that even minor changes in the composition of protein-rich fluids like plasma and serum may have a significant impact on the formation of the PC on the surface of NPs [62].

The properties and potential interactions of nanomedicines with proteins in BFs are significantly affected by their physico-chemical properties, including size and surface properties [63]. In the following sections, we discuss the relevant properties of NPs that may influence the formation of PCs and thus their biological activity as potential cancer therapeutics.

### 3.1. Size

The size of NPs is one of the dominant factors that has a key role in protein adsorption, conformational changes, and PC composition, and thereby biological interactions. Typically, larger NPs exhibit a greater ability to adsorb proteins compared to their smaller counterparts due to their higher curvature suppressing protein adsorption [64]. Previous research has shown that a slight change of 10 nanometers in the size of silica NPs can significantly impact the composition of polycarbonate. Small silica NPs revealed a much higher attachment to lipoprotein clustsacaerin, while larger silica NPs absorbing more prothrombin or the actin regulatory protein gelsolin [65]. Similarly, another study revealed that the thickness of the adsorbed protein layer progressively increased with the size of gold NPs incubated with human blood [66].

However, it has been also shown that PC formation on poly(lactic-co-glycolic acid) (PLGA) and PLGA-PEG NPs does not exhibit a significant correlation with NP size [67]. In general, the difference in the PC composition of NPs with similar sizes can be attributed to several factors, including the surface charge of NPs, the NP material, and the analytical methods applied [68].

### 3.2. Shape

The shapes of NPs play a significant role in determining the type and amount of proteins adsorbed onto their surfaces. Deng et al. [69] discovered that in addition to albumins and apolipoproteins, TiO_2_ nanorods were mainly adsorbed IgM and IgG proteins, whereas fibrinogen was the major protein attached to the nanotubes. In the case of gold NPs, spherical and rod-shaped Au NPs had a higher amount of total PC adsorption than caged Au NPs, as the caged structure had low curvature areas and sparse ligation on flat surfaces [70]. Similarly, García-Alvarez et al. [71] found that gold nanostars bind a greater amount of protein than gold nanorods of the same size, which may be attributed to an increase in specific surface area.

In another study, it has been demonstrated that rod-shaped mesoporous silica NPs have a significantly larger amount of protein attached to them than globular silica NPs, regardless of their identical chemistries, porosities, surface potentials, and sizes [72]. Furthermore, it has been reported that the morphology of NPs is able to significantly influence their efficacy in various aspects such as cellular uptake, duration in the bloodstream, internalization by immune cells, distribution throughout the body, and duration of residence within cells [73]. Studies with Hela and Caco-2 cells have demonstrated that rod-shaped particles are internalized more efficiently in both cell lines [74,75]. These studies indicate the possibility of manipulating NP morphology as a tool to engineer the next generation of drug carriers and theranostic vehicles.

### 3.3. Charge and Chemistry

The surface charge of NPs is a crucial determinant in regulating the composition, binding affinity, and structural alterations, thereby potentially influencing protein–protein interactions and the biological functionality of NP [63].

Under normal physiological conditions, proteins in plasma typically carry a negative charge. These proteins are also the most abundant ones found on NPs [76]. Positively charged NPs are often more prone to clearance by the reticuloendothelial system (RES) because they are easily recognized by opsonins. Notably, negatively charged NPs are also capable of protein adsorption, partly because of the presence of cationic species in BFs [77]. Interestingly, the opsonization process can be prevented by coating NPs’ surfaces with negatively charged groups, which eventually may increase their stability in BFs [78].

Gessner et al. [79] made the significant finding that a rise in charge density correlated with an elevated quantity of proteins that were assimilated onto the polymer NP’s surface. They found that tannic acid-modified gold NPs adsorb more proteins than PEG-modified gold NPs. The higher concentration of negative charges on the tannic acid-modified gold NPs compared to the PEG-modified gold NPs is responsible for enhancing the electrostatic interactions between the NPs and proteins. Similarly, Lai et al. [80] found that positively charged PEGylated polyethylamine glycol gold NPs absorbed 2–4 times more proteins than negatively charged citrate glycol gold NPs.

Moreover, the internalization pathways of NPs into cells can also be influenced by their surface charge. For example, a study demonstrated that quantum dot NPs featuring a negative surface modification were absorbed more effectively into HEK cells through the lipid raft/caveolae pathway [81]. In contrast, Harush-Frenkel et al. [82] showed that the positively charged NPs, unlike the negatively charged ones, have a higher rate of endocytosis utilizing the clathrin-mediated endocytosis pathway. Additionally, positively charged NPs are more likely to adsorb proteins with pI <5.5, such as apolipoproteins and complement proteins, whereas negatively charged NPs have a greater affinity for pro-teins with pI >5.5, such as fibrinogen and hemoglobin [83]. Tenzer et al. [84] and Hata et al. [85] demonstrated that the surface PCs of unmodified silica NPs and carboxyl- and amino-modified silica NPs exhibited different compositions. NPs with an amino modification have a positive charge and adsorb less albumin (pI 5.92), transferrin (pI 6.81), coagulation factor II, and IgG (pI 6.4–6.9) compared to unmodified and carboxyl-modified NPs; thus, they are less opsonized and exhibit higher stability. Nevertheless, the involvement of surface charge in PC formation is not consistent across all cases as additional forces play a role in the process. Sakulkhu et al. showed that glycoproteins adsorbed to the NPs and formed PCs in spite of the surface charge of iron oxide NPs [86].

In conclusion, these findings provide important evidence for the design of an ‘optimal NP surface’ for cancer therapy. Important surface modifications of NPs aiming at the improvement of biocompatibility, targeting ability, and therapeutic efficacy are shown in Figure 2.

## 4. Specialized Nanostructures for Cancer Treatment

Cancer remains a leading factor in hospital admissions and fatalities, with approximately 8.8 million deaths and 15.2 million new diagnoses reported in 2015. Projections suggest that there will be 28 million new instances of cancer annually across the globe by the year 2040 if incidence, population growth, and ageing remain stable [87]. Due to ongoing research, treatment options and patient outcomes have improved within the recent decade. However, there are still various challenges hampering the effective treatment of malignancies, such as late diagnosis (mostly in advanced stages), patient and cancer heterogeneity, therapy resistance, and complications. The application of nanotechnology-based tools presents an opportunity to address these challenges; thus, they have the potential to improve cancer diagnosis, disease monitoring, and patient outcomes by utilizing targeted therapies. NPs are already utilized in a wide variety of biomedical implementations, for instance, as contrast agents for magnetic resonance imaging (MRI), in therapy as DD vehicles, and in combination with certain antibodies on the surface of the NP to label certain cell populations. The interest in nanostructured drugs has increased within the recent decade and was represented by a growing number of original publications analyzing nanostructured tools for cancer treatment (Figure 3). In particular, NMs are investigated for the treatment of common cancers, namely breast, colon, and prostate cancer [88]. In contrast, there is a lack of nanomaterial-based research aimed at the treatment of other cancers, such as head and neck cancers (HNCs) (Figure 3, number of publications indicated in red below). HNCs are among the top ten of cancer types worldwide and still exhibit poor survival rates, which have not significantly improved within the recent decade [89]. Due to the development of therapy resistance and associated subsequent near or distant relapses, there is a high need for novel treatment options, including NP-based therapy.

Since the early stages of nanotechnology research, studies were conducted in HNC models and aimed at overcoming the lack of conventional chemotherapeutic agents’ specificity, such as cisplatin [90]. The majority of research investigations are currently situated in the preclinical phase; however, they have unveiled encouraging possibilities for enhancing the efficacy of cancer treatment. However, de Lima et al., in a systematic review, already reported a significant lack of these achievements being able to be translated to NP-based chemotherapy formulations to clinically benefit patients with HNCs [90].

In the following section, we would like to provide an introduction of general strategies of nanoparticulate drug systems for cancer therapy. By focusing on innovative approaches of recent studies, we want to provide a new impetus, especially for the treatment of head and neck tumors. For a more general analysis of NP-based drugs and their molecular mechanisms, please refer to other reviews on this topic [91,92,93].

### Applications

Due to their unique biophysical properties, NMs exhibit promising tools for cancer treatment, allowing passive accumulation of anti-tumor agents at the tumor site as well as active targeting of cancer cells. Additionally, nanoformulated drugs are able to pass physiological barriers, such as the blood–brain barrier (BBB) or the stomach epithelium, and thus have clear advantages over many conventional compounds. For the challenging treatment of brain tumors, researchers designed advanced NP-based systems allowing that can pass the BBB and thus deliver drugs to tumor cells within the brain [94]. Key NPs for cancer applications include nanoliposomes, nanoemulsions, polymeric NPs, quantum dots, and gold, iron, and carbon NPs [95]. In addition to general advantageous properties, each NP class possesses specific features that can be utilized for applications in nanomedicine (Figure 4). Liposomes and emulsions show ideal membrane delivery properties, resulting in decreased drug toxicity and increased efficacy [96]. On the other hand, polymeric NP are used for their prolonged circulation time [97], and quantum dots are versatile bioimaging tools, especially for cancer imaging [98]. The application of inorganic NPs, such as gold, iron, and carbon NPs, are manifold, including all fields of nanomedicine. The advantageous features of gold NPs encompass high chemical stability and thus biocompatibility, as well as high tunability. In addition to the previously mentioned features, iron NPs also exhibit unique magnetic properties, which make them particularly suitable for use as contrast agents in imaging procedures [99]. Due to their high degree of biodegradability and biocompatibility, as well as their diversity, peptide-based NPs offer versatile tools for biomedical applications, such as tissue engineering, regenerative medicine, bioimaging, antibacterial applications, and (of course) cancer treatment [100].

The major strategy of applying nanoparticular formulations of chemotherapeutics for passive targeting of cancer cells is mainly based on the so-called EPR effect. This effect de-scribes passive accumulation and thus enhances the cellular entry of the chemotherapeutic substance into the tumor area. Due to vascular abnormalities of newly formed blood vessels in fast-growing tumors, as well as impaired lymphatic drainage of the tumor tissue, NPs are able to extravagate from blood vessels into the tumor tissue [101] (Figure 5A). The EPR effect is therefore one strategy to distinguish the treatment of diseased from healthy cells and discuss the possible cancer imaging and diagnosis techniques that could be used [102]. However, practical applications show that therapeutics only exhibit limited success, purely based on the EPR effect. A meta-study retrospectively analyzing literature from ten years could show that typically only 0.7% of total given NP formulation reaches the target side [103]. Nevertheless, it is discussed that NP drugs might still benefit from the EPR effect and bring real advantages for patient care, given the right disease indication and patient stratification.

One of the biggest success stories of NPs in cancer therapy up until now is Abraxane. Abraxane contains nanoformulated paclitaxel in a human serum albumin (HSA) shell and was introduced as the first ever NP drug in 2005 after obtaining approval from the FDA in the US. Its indications include breast cancer, lung cancer, and pancreatic cancers [104]. It has also been reported in the literature that NP albumin-bound paclitaxel provides more effective paclitaxel delivery to tumor cells with fewer side effects than traditional chem-otherapy strategies [105]. Clinical studies showed evidence for reduced systematic toxicity of paclitaxel nanoformulations, but they often lack in proper study design [90].

Besides Abraxane, there are also other nanoformulated drugs with various characteristics approved by the FDA. Within the group of metallic NPs, Feraheme, a drug containing an iron oxide NP formulation, is approved for the treatment of metastatic prostate and testicular cancer [91]. Further examples are Doxil (liposomal doxorubicin), Eligard (poly (DL-lactide-coglycolide)), DaunoXome (liposomal daunorubicin), or Onivyde (liposomal irinotecan), which are approved for the treatment of different solid cancers or leukemias (for more detail, see [91]).

Due to the low efficiency of EPR-based drug targeting [102], nanotechnical methods for active tumor targeting are increasingly becoming the focus of research. Active tumor targeting involves the functionalization of NPs with ligands that specifically bind to receptors overexpressed on cancer cells, facilitating their selective accumulation within tumors (Figure 5B). This targeted delivery enhances drug efficacy and reduces systemic toxicity. Additionally, so-called “smart NPs” are engineered with sophisticated functionalities that enable them to respond dynamically to cues in their microenvironment or external stimuli [106]. Such responsiveness allows for on-demand drug release, triggered by factors like pH, temperature, or specific biomolecules present in the targeted tissue (Figure 5C). Furthermore, smart NPs can be designed with diagnostic capabilities, enabling real-time monitoring of treatment efficacy or disease progression.

Regarding the fact that therapy resistance and associated subsequent near or distant relapses are common in patients with HNCs [107], the application of smart NPs can especially offer a promising approach to overcome resistance. For breast cancer, several studies have already described novel nanotheranostic systems, which can prevent tumor recurrences [108]. These smart NPs in combination with chemo- and/or immunotherapy have the potential not only to kill tumor cells but can also help to overcome radio- and chemoresistance mediated by cancer stem cells. For example, doxorubicin-loaded pH-sensitive mesoporous silica NPs showed high sensitivity to a low pH as well as high cellular uptake efficiency compared to free doxorubicin [109]. Recent studies also analyzed the applicability of smart NPs for head and neck squamous cell carcinoma (HNSCC). For example, Zhang et al. designed a dual-stimuli-triggered smart nanoprobe based on Fe^3+^, polyphenol tannic acid, and a hyaluronic acid matrix (Fe^III^TA@HA) [110]. Specific targeting of HNSCC cells is achieved by CD44 overexpression, and the abundance of enzymatic activity in the tumor microenvironment (TME) triggers the release of the Fe^III^TA complex, finally inducing the ferroptosis/apoptosis of tumor cells [110]. Additionally, these nanoprobe exhibited good biosafety, suggesting that these results can be clinical translated.

Tumor-targeting NPs that are responsive to changes in pH and can convert their charge have been employed to enhance the efficacy of chemo- and gene therapeutics in the treatment of HNC by Lo et al. [111]. These NPs have the potential to minimize side effects while improving the active targeting of tumors and facilitating the delivery of therapeutics to the nucleus and cytoplasm. Oxaliplatin (Oxa) is a platinum compound of the third generation that hinders the process of DNA replication. The regulation of cancer cell apoptosis, resistance, and progression may be influenced by miR-320. To enhance the efficacy of miR-320 and Oxa, novel NPs were developed. These NPs were engineered with a ligand, a cell-penetrating peptide, and a nucleus-targeted peptide. NPs were enveloped with a shield that could be adjusted for charge and size in order to protect peptides from degradation. This shield was removed at acidic tumor sites, allowing the peptides to be exposed for targeted delivery. The findings showed that the NPs, as designed, had consistent sizes and high drug encapsulation rates. The pH-responsive release and uptake of Oxa and miR-320 into human tongue squamous carcinoma SAS cells were demonstrated by the NPs. Effective delivery of Oxa and miR-320 to the nucleus and cytoplasm, respectively, was achieved by these NPs. This study is the first one to showcase the simultaneous intracellular adjustment of the NRP1/Rac1, PI3K/Akt/mTOR, GSK-3β/FOXM1/β-catenin, P-gp/MRPs, KRAS/Erk/Oct4/Yap1, and N-cadherin/Vimentin/Slug pathways in order to impede the proliferation, advancement, and resistance to multiple drugs in cancerous cells. In mice that carry SAS, the administration of NPs loaded with Oxa and miR-320 together showed enhanced effectiveness in fighting tumors and significantly reduced the toxic effects associated with Oxa. These NPs, which are localized in the nucleus and cytoplasm, have a coating that is sensitive to the pH of the tumor and can be adjusted in terms of size and charge. This innovative nanoplatform has the potential to serve as a valuable combination therapy for delivering nucleic acids and chemotherapeutic agents, ensuring optimal safety and efficacy in the treatment of HNC.

The conventional utilization of concurrent chemoradiotherapy in the treatment of lo-co-regionally advanced unresectable head and neck squamous cell carcinoma (HNSCC) has limited therapeutic benefits due to the toxic side effects of systemic clinical radio-sensitizers. The delivery of these drugs through regional platforms is challenging because of limited drug permeation. However, it offers the advantage of spatio-temporal control of combinatorial regimens locally, which can help overcome drug resistance. Bhardwaj et al. [112] have addressed these challenges by developing biodegradable gellan- and lipid-based dual nanocarriers-in-ion-triggered porous mucoadhesive hydrogels. These nanocarriers are designed to enhance the site-specific delivery of clinically relevant radiosensitizers, such as cisplatin and paclitaxel. Interestingly, the use of NP-in-gel formulations resulted in prolonged tumor bioaccumulation of both chemotherapeutic drugs, while reducing systemic absorption. This improved the in vivo efficacy of the treatment, as confirmed by PET-CT imaging, and enhanced safety compared to systemic commercial formulations approved for HNSCC chemoradiotherapy. The NPs facilitated the uptake of radiosensitizers into cancer cells, leading to cell arrest and a synergistic enhancement of radiation-induced DNA damage and apoptosis. These findings suggest that the developed platform has clinical potential for the loco-regional management of HNSCC requiring curative chemoradiotherapy [112].

While addressing the problem of chemoresistance, especially for metastasized HNSCC, Zhou et al. designed MnO_2_-based nanoshells that were loaded with the chemotherapy agents docetaxel and cisplatin [113]. These smart nanoshells responded to an acidic pH of the TME enabling tumor-specific imaging (MRI) and the efficient release of the encapsulated drug. Furthermore, this nanosystem attenuated hypoxia of the tumor, and thus showed a synergistic therapeutic effect in vitro and in vivo [113].

Radiotherapy is a primary treatment modality for patients with HNC, but its efficacy is limited by radiation-induced side effects and radioresistance. Various radiosensitizing strategies are being explored to enhance treatment outcomes while minimizing toxicity. RNA therapeutics show promise as radiosensitizers by targeting specific pathways as-sociated with radioresistance. Nevertheless, challenges in clinical translation persist, primarily due to delivery obstacles. Dehghankelishadi et al. [114] introduced high-density lipoprotein NPs (HDL NPs) as biocompatible carriers for miR-34a, a well-known radiosensitizing RNA. The researchers utilized a microfluidic-based method to efficiently produce miR-34a-loaded HDL NPs. Evaluation of the radiation response in UM-SCC-1 head and neck cancer cells revealed reduced metabolic activity and increased radiation-induced apoptosis following treatment with miR-34a-HDL NPs. Furthermore, the radiosensitizing effects of miR-34a-HDL NPs were validated in a physiologically relevant co-culture spheroid model of head and neck cancer, demonstrating enhanced apoptotic activity and cell cycle disruption. These findings underscore the potential of HDL NPs as an effective delivery platform for radiosensitizing RNA, as evidenced by the augmented radiobiological responses observed in both 2D and 3D models [114].

Delavarian et al. [115] conducted a clinical study to examine the impact of oral curcumin on radiation-induced oral mucositis (OM) in individuals diagnosed with head and neck cancer. The primary objective of this investigation was to assess the efficacy of oral curcumin in managing OM. To enhance the solubility, absorption, and bioavailability of curcumin in aqueous solutions, the researchers employed curcumin nanomicelles. These NPs were then evaluated for their clinical effectiveness in treating OM. The study involved 32 patients diagnosed with head and neck cancer who were divided into two groups: a case group and a control group. During the course of radiotherapy, the case group received nanocurcumin, while the control group received a placebo. The severity of mucositis was assessed at regular intervals, and the researchers observed a statistically significant difference in mucositis severity between the two groups throughout the study period. Interestingly, the researchers noted that only 32% of the case group developed oral mucositis compared to the control group, where all patients developed it by the second week of radiotherapy. Furthermore, the administration of nanocurcumin did not result in any significant oral or systemic side effects. These findings indicate that nanomicelle curcumin is an effective agent in preventing or reducing the severity of oral mucositis. Based on these results, the researchers suggested that the application of nanocurcumin may be a viable approach to prevent the development of oral mucositis in patients with HNC undergoing radiotherapy. This study contributes valuable insights to the existing literature on the potential benefits of oral curcumin in managing radiation-induced OM [115]. Nanomicelles, as a significant class of nanostructures, offer an alternative approach that can enhance the antitumor effect by extending the duration of action of chemotherapeutics and facilitating greater drug accumulation at the specific site [116].

Nanomicelles have the potential to revolutionize the treatment of HNC by enabling innovative approaches that involve delivering chemotherapeutic drugs simultaneously. This co-delivery strategy aims to enhance the antitumor effectiveness of these drugs by exerting more potent cytotoxic effects on cancer cells, surpassing the efficacy of free drugs [117]. For example, the combination of methotrexate (MTX) and salinomycin (SAL) was examined by Zhu et al. [118]. In their study, MTX was linked to the hydrophilic outer shell of the DD system because it had a strong attraction to the overexpressed folate acid receptors found in tumor cells. On the other hand, SAL was enclosed within the hydrophobic inner core of the system, allowing it to be released specifically in the central areas of solid tumors However, there are certain drawbacks associated with the utilization of nanomicelles in DD applications. These limitations include the premature release of drugs, inadequate control over sustained release, and the incapability to encapsulate hydrophilic agents [119].

Mapanao et al. [120] utilized nonpersistent ultrasmall nanogold structures to implement a combined chemo-photothermal treatment for HNSCCs. These nanostructures are composed of excretable narrow near-infrared (NIR) absorbing gold ultrasmall particles and an endogenously dual-controlled cisplatin prodrug. The efficacy of these nanostructures was assessed using three-dimensional (3D) models of HNSCCs with either a positive or negative human papillomavirus (HPV) status. The combined therapy exhibited a more significant antitumor impact on HPV-positive HNSCCs. In conclusion, the results highlight the potential clinical significance of adaptable noble metal-based combined therapies in tumor treatment.

A common complication of HNSCC in the oral cavity is the invasion of tumor cells to jaw bones. Due to the limited access of conventional chemotherapeutic drugs to bone tissue, the application of smart NPs is also promising here. Chen et al. engineered a nano-biomimetic DD vehicle which does not only target bone and cancer tissue but also shows immune escape ability [121]. Here, tumor toxicity can be triggered by hypoxia at the tumor site during a photothermal treatment process [121].

Besides the established treatment of HNC based on (radio)chemotherapy, combined treatment schedules including immune checkpoint inhibitors, such as pembrolizumab, have revealed improved prognosis in clinical studies, offering new hope to patients with advanced or recurrent disease [122]. Zhou et al. have addressed this shift of HNC therapy towards immunotherapy in a recent study [123]. The authors designed an extracellular vesicle-based intelligent nanoplatform, which is able to respond to and modulate the tumor microenvironment, thereby inducing immunogenic cell death. Here, the use of extracellular vesicles that mimic natural occurring exosomes offer significant and distinctive advantages over synthetic NPs, such as good cell delivery properties and reduced immune response.

The use of smart NPs represents a cutting-edge approach in nanomedicine, offering unparalleled potential for precise and tailored therapeutic interventions. Recent studies revealed versatile nano tools to specifically target (head and neck) cancer cells. However, challenges persist, including the complexity of NP synthesis, potential toxicity concerns, and the risk of immune system recognition. Despite these drawbacks, the use of NPs holds great promise in revolutionizing cancer treatment paradigms, offering the potential for more effective and personalized therapies.

## 5. Shortcomings, Current Trends, and Future Perspectives

Nanomaterials (NMs) are substances that consist of particles, of which at least half have dimensions ranging from 1 to 100 nm. Various classifications have been established to categorize NMs based on their size (1D, 2D, or 3D), origin (synthetic or natural), shape (high or low aspect ratio), and composition (organic, inorganic, composite/hybrid, and carbon) [124,125]. The size of NMs plays a significant role in determining their applications. For instance, NMs with a size smaller than 20 nm are commonly employed in detection and imaging processes because of their ability to penetrate the BBB and be efficiently eliminated through the kidneys. On the other hand, NMs larger than 20 nm are frequently utilized as carrier systems as they are able to easily bypass physiological barriers and remain in circulation for extended periods. The ability to manipulate the sizes, shapes, morphologied, surface charges, and physicochemical properties of NMs allows them to serve as effective carrier systems for delivering drugs, genes, and vaccines [126]. Enhanced pharmacokinetic characteristics and biodistribution of substances delivered via nanostructures have the potential to facilitate their extensive application in the field of biomedicine, particularly in the realms of tumor detection and therapy [127].

A range of treatment modalities, including chemotherapy, gene therapy, immunotherapy, photodynamic therapy, and combination therapy, have been employed in the management, mitigation, and eradication of cancer. Nevertheless, the successful and safe transport of therapeutic substances to tumor sites encounters numerous obstacles, such as unwanted degradation, immune system elimination, various biological barriers, and harmful side effects. Here, NMs present a promising avenue for overcoming these challenges. Tumor microenvironments (TMEs) exhibit distinct characteristics, such as metabolic suppression, metabolic reprogramming, and slight acidity, representing promising targets for therapeutic interventions [128]. Consequently, future studies should further explore the potential of TME-responsive nanostructures and their diverse utility in cancer treatment.

There has been a notable increase in research efforts in recent years focusing on the advancement of DNA-based DDSs [129]. DNA nanostructures have garnered significant attention in the field of cancer nanomedicines due to their ability to be precisely controlled in terms of size and shape. These nanostructures offer excellent biocompatibility, programmability, and biodegradability, making them highly desirable for use in medical applications. Additionally, DNA nanostructures can be easily functionalized, further enhancing their potential for targeted DD and therapy. The integration of aptamers into DNA nanostructures has resulted in the development of efficient instruments for bioimaging and biosensing, in addition to targeted cancer treatment. Molecular recognition possesses exceptional properties that facilitate the creation of diverse nanostructures. The efficient cellular uptake and effective drug encapsulation capabilities highlight DNA origami as a perfect “smart” core element with customizable potential for the innovation and production of adaptable, secure DDs. DNA origami structures exhibit biocompatibility and biodegradability, allowing for functionalization with various components, including aptamers, lipids, proteins, and inorganic NMs [130]. DNA origami, characterized by a simple synthesis process and excellent biocompatibility, has found extensive applications in various biomedical fields, including cancer treatment, imaging, and diagnosis [131].

Apoptosis induction in cancer cells is the main approach in the management of the majority of cancers. Regrettably, the development of drug resistance in cancer cells has resulted in a notable reduction in treatment efficacy, which holds great significance for individuals battling cancer. Hence, novel therapeutic approaches and tactics are required to address clinical demands. Metal ion chelation therapy, as an innovative treatment modality, has yielded several research findings in the realm of cancer therapy. Zinc ions are essential for cell metabolism and growth, acting as a ubiquitous coenzyme factor in various cells. Nevertheless, if the intracellular concentration of Zn^2+^ surpasses the cell’s tolerance threshold, it can harm intracellular mitochondria and trigger an outburst of reactive oxygen species (ROS), resulting in severe intracellular oxidative stress known as “Zn^2+^ interference”. The use of Zn-NPs as a novel “Zn^2+^ intervention” therapy is promising due to its remarkable therapeutic efficacy and exceptional safety profile [132]. It is quite possible that new/similar methods will contribute to the literature with the discovery of the effect of metal ion interference.

NPs have seen a rise in utilization over the recent ten years, particularly in the realm of diagnosing and treating specific forms of cancer. Due to their ability to transport a range of substances, such as imaging agents, medications, genes, vaccines, radiosensitizers, and photosensitizers, NMs are pivotal in advancing innovative technologies for managing certain cancer types. Indeed, studies have shown that NMs enhance delivery effectiveness, enhance patient outcomes, and enable targeted delivery, controlled release, responsiveness to stimuli, and multi-agent delivery. With the rise in patient numbers and the shortcomings of conventional treatment approaches, there is a pressing need for innovative methods in the detection and management of cancer/various forms of cancer. In recent years, substantial advancements have been achieved in the development, creation, and manufacturing of NMs that offer distinct benefits for diagnosing and treating tumors. This progress is evident from the notable surge in publications related to ‘nanomaterials and tumor/cancer’ in PubMed/Embase/SCI. Ongoing investigations in this area are introducing novel strategies to address the limitations of traditional techniques utilized in tumor diagnosis and treatment [127].

Nanostructures are anticipated to have an additional impact on the efficacy of conventional chemotherapy and radiotherapy, particularly in terms of enhancing the healing process. The utilization of nanostructures can lead to a significant improvement in this regard. While nanostructures can passively deliver targeted treatment through the EPR effect, their combination with aptamers, receptor-specific peptides, and monoclonal antibodies can actively facilitate targeted delivery. By employing this approach, nanostructures have the potential to enhance the effectiveness of traditional chemotherapy and radiotherapy, resulting in optimized tumor eradication, minimized side effects associated with dosage, and reduced occurrence of severe complications [133,134].

In the future, it is anticipated that nanostructures will have the ability to perform multiple functions simultaneously. These nanostructures possess the capability to respond to various stimuli, enabling them to aid in tumor diagnosis and be utilized in combined treatment approaches, like photodynamic therapy (PDT) and photothermal therapy (PTT) [135]. Consequently, these nanostructures can assist in regulating the timing and intensity of treatments, leading to a reduction in adverse reactions and an enhancement in the efficacy of cancer therapy.

Most nanocarrier systems are used for tumor treatment via intravenous injection and systemic administration with the ability to selectively aggregate around tumors through defective tumor microvessels. However, the EPR effect is still not fully understood, and the safety of NMs remains an important issue [136,137,138].

Polymeric nanoplatforms continue to be utilized in cancer therapy; however, achieving precise control over drug release remains a challenge. To enhance treatment efficacy and minimize side effects, polymeric nanoplatforms must fulfill specific criteria. These nanoplatforms are engineered to release therapeutic agents consistently and selectively at targeted sites in response to various stimuli, such as pH, temperature, enzymes, among others. The ability to respond to external stimuli enhances the efficiency of nanocarriers, facilitating improved DD to diseased tissues [34]. These nanostructures are extremely promising for the release of cancer drugs in the future.

Peptides and proteins have been of interest for functional biomedical materials due to their widespread occurrence in nature, easy accessibility, and biocompatibility [139,140]. Peptide-based nanostructures undergo self-assembly, leading to the creation of diverse nanomorphologies. This enhances the stability of the resulting molecules, simplifies the synthesis process, and reduces costs. Especially, short peptides such as dipeptides are anticipated to gain popularity as materials for self-assembly due to their adaptable nature. Additionally, the remarkable biocompatibility and biological functionality of peptide-based NMs highlight their potential for extensive use across different sectors, particularly in the realm of cancer treatment [100].

The drug-loading efficiency of nanocarriers, which refers to the proportion of drug in the overall particle mass, is greatly influenced by the characteristics of the particular nanocarrier and drug. Presently, the majority of NPs demonstrate relatively low drug-loading efficiencies, but there are ongoing efforts to develop new and enhanced methodologies. Protein-based nanodelivery systems have the capability to encapsulate drugs within hydrophobic regions of proteins or between various protein components. Protein-based nanostructures exhibit a highly promising potential for future research due to their exceptional drug-loading capacity, which can reach up to 90% [141].

Despite the fact that in vitro preclinical data have demonstrated the pharmacological characteristics of nanocarriers, including drug concentration, clearance rate, and half-life, accurately monitoring the journey of drugs from their introduction into the human body to their elimination remains a challenge. Furthermore, the disparity between basic preclinical research data and clinical trial data can be attributed to the absence of tumor models capable of accurately simulating human tumors. This limitation hinders our comprehension of the intricate interaction between NMs and complex organisms. Hence, it is essential to create tumor models, such as humanized animal models, that can effectively simulate tumor heterogeneity and microenvironment within the human body in order to facilitate the clinical application of nanotechnology. Additionally, a frequently neglected yet equally significant concern is the reduction in the nanomaterial load following systemic delivery as it travels through the circulatory system towards tumor sites. Consequently, the release of drugs must be taken into account during the clinical translation of nanocarriers. Various stimuli-responsive NMs have been developed to control drug release [126,142,143]. These designs are expected to have promising results for the future.

Exosomes, which are small extracellular vesicles measuring approximately 100 nm, are released by most cells and contain a variety of bioactive molecules, such as nucleic acids, proteins, and lipids. One of the key advantages of exosomes is their high biocompatibility and low immunogenicity, making them ideal for delivering therapeutic agents like chemotherapeutic drugs, nucleic acids, and proteins [144]. Notably, both hydrophilic and hydrophobic chemotherapeutic drugs, including Dox and PTX, have been successfully loaded into exosomes. Numerous studies have demonstrated that the exosome-mediated delivery of chemotherapeutic agents can significantly enhance the anti-cancer effects of these drugs [145,146,147,148,149]. Exosomes have revolutionized the field of DD owing to their remarkable attributes, such as low immunogenicity and excellent biocompatibility. Traditional methods employed for delivering anticancer agents, nucleic acids, and proteins in cancer treatment often fall short in achieving the desired therapeutic effects due to the degradation of the therapeutic agent within the body and the absence of targeted deliveries. By harnessing the inherent advantageous properties of exosomes, numerous investigations have substantiated their potential as carriers for drugs or as engineered entities for anticancer therapy. Despite the existence of several challenges and obstacles in establishing a commercial exosome-based DDSs, a comprehensive comprehension of the intricate biological mechanisms of exosomes, coupled with further clinical studies, will pave the way for their emergence as a cutting-edge nanoplatform for cancer therapy [144].

## 6. Conclusions

Nanotechnology possesses the capability to revolutionize the treatment of cancer through the utilization of nanomaterials (NMs) for precise drug delivery and therapy. Nevertheless, numerous obstacles remain in enhancing the clinical application of NMs. Important aspects of limitations in NP-based therapies include (1) their biocompatibility and toxicity: some NPs may cause immune reactions or toxicity in the body, limiting their clinical application. The long-term effects of NPs on human health are still not fully understood. (2) Off-target effects: despite their targeted delivery capabilities, NPs may still accumulate in healthy tissues, leading to off-target effects and potential toxicity. (3) Biodegradability: many NPs have limited biodegradability, which may result in their accumulation in the body over time, potentially causing long-term adverse effects. (4) Cost and scalability: The production of NPs for clinical use can be expensive, and large-scale production may be challenging. This can limit their widespread use in cancer treatment. (5) Complexity of design and characterization: designing NPs with the desired properties for targeted drug delivery can be complex, and characterizing their structure and function is often challenging. (6) Drug loading and release efficiency: ensuring efficient loading of drugs onto NPs and controlling their release kinetics can be difficult, affecting the efficacy of the treatment. (7) Resistance: cancer cells may also develop resistance to NP-based therapies over time, reducing their effectiveness.

Addressing these limitations requires further research and development to optimize the design, synthesis, and application of NPs in cancer treatment [150].

Conversely, these findings also allow for conclusions to be drawn regarding the optimal properties of NPs. Optimal physicochemical properties may vary depending on the specific therapeutic application and the targeted tissue or cell type. NPs with a small size [2], neutral or slightly negative surface charge [151], appropriate surface functionalization, shape, surface coating, drug loading, and release capabilities, and high biodegradability and biocompatibility are believed to be optimal for efficient therapy (Figure 6).

Despite the notable progress made in cancer treatment, ongoing research and advancements in nanotechnology remain crucial. Achieving direct and efficient treatment approaches necessitates the collaborative endeavors of multidisciplinary teams comprising scientists, clinicians, institutions, and industrial stakeholders. Nanotechnology also offers the potential to tailor treatments to individual patients by considering the unique molecular properties of tumors. These promises not only have the potential to increase effectiveness but also to reduce side effects, thus improving the quality of life of patients with cancer. However, despite the great promises of nanotechnology, there are still challenges in cancer treatment, and efforts are continuously being made to overcome the difficulties in effective DD. While the potential of improving cancer detection and treatment through nanotechnology is exciting, the journey from translating the laboratory setting into a clinical routine is challenging. It is the shared duty of the scientific community to actively work towards transforming this promise into a tangible reality.

## Figures and Tables

**Figure 1 ijms-25-05171-f001:**
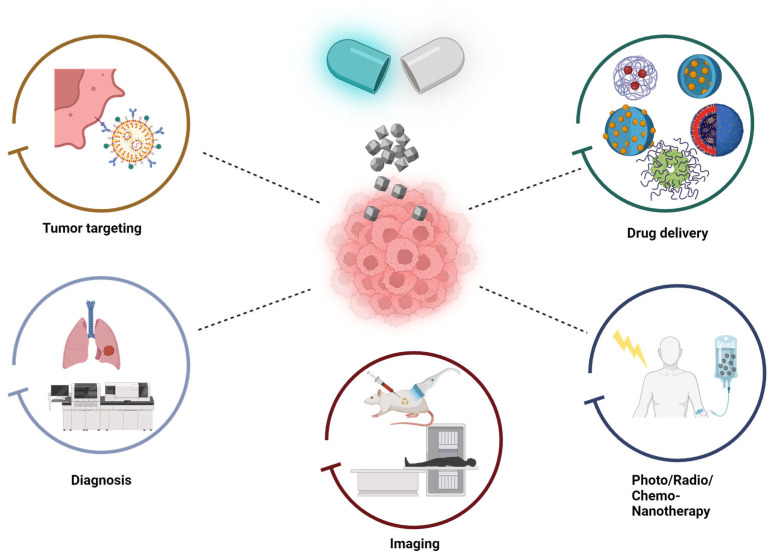
Exploring nanoparticles (NPs) for cancer therapy. NPs are utilized for various purposes in cancer treatment, including tumor targeting, drug delivery, photo-/radio-/chemo-nanotherapy—where NPs act as active agents, not just as carriers—as well as imaging and diagnosis. Created with BioRender.com.

**Figure 2 ijms-25-05171-f002:**
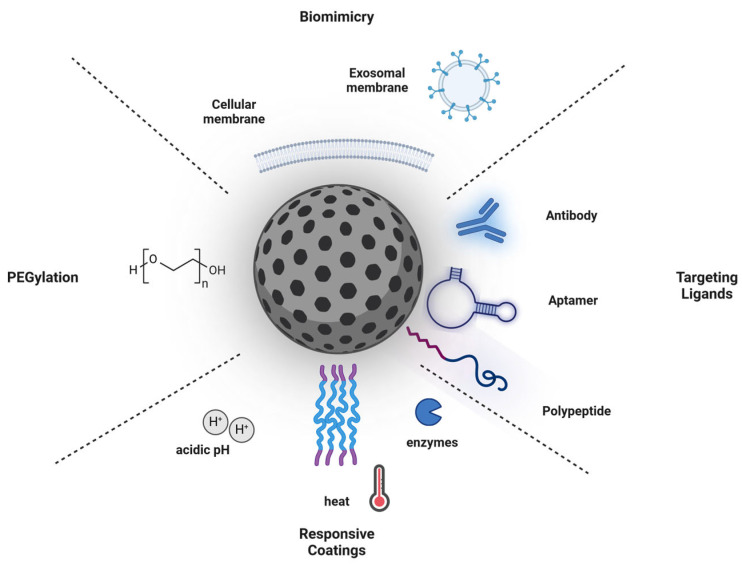
Surface modifications of nanoparticles for cancer therapy. Surface modifications of NPs aiming at the improvement of biocompatibility, targeting ability, and therapeutic efficacy can be classified into four groups: (1) Biomimetic coatings, which are mainly made of different (bio)membranes mimicking a biological identity. (2) Targeting ligands: NPs are conjugated with targeting ligands such as antibodies, peptides, or aptamers to enhance specific binding to cancer cells or tumor vasculature. (3) Responsive coatings, which include different polymers that are sensitive towards an acidic pH or enzyme that is present in the tumor microenvironment, or external heat, resulting in controlled drug release. (4) Polyethylene glycol (PEG)ylation of NPs improves their stability, reduces clearance by the immune system, and prolongs circulation time in the bloodstream. Created with BioRender.com.

**Figure 3 ijms-25-05171-f003:**
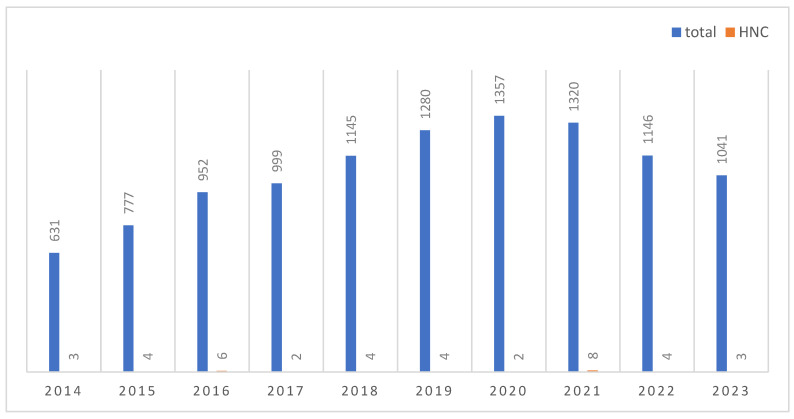
Studies analyzing nanostructured tools for head and neck cancer treatment are significantly underrepresented among all studies dealing with NP-based cancer applications. Blue: number of total publications by publication year (Pubmed search query: “cancer [title] treatment nanostructure nanoparticle NOT review”; accessed on 1 March 2024). Red: number of HNC-related publications by publication year (Pubmed search query: “head and neck cancer [title] treatment nanostructure nanoparticle NOT review”; accessed on 1 March 2024).

**Figure 4 ijms-25-05171-f004:**
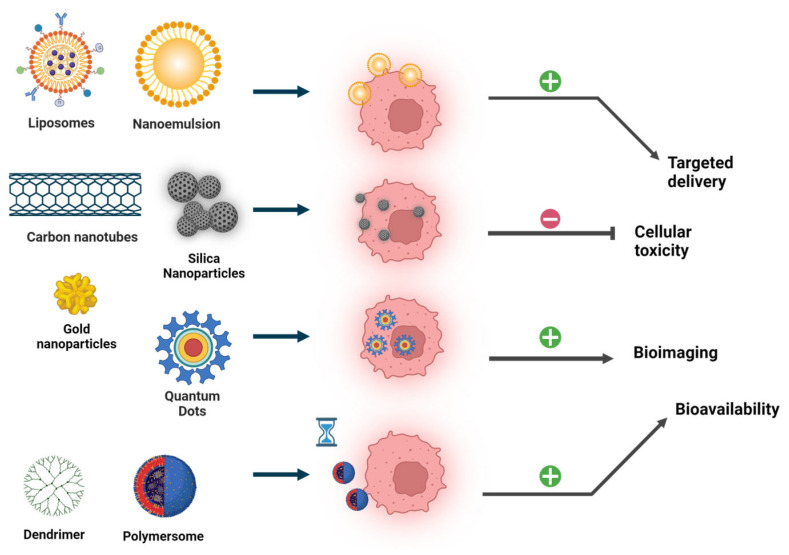
Advantages of nanoparticles (NPs) in cancer treatment. NPs commonly used for cancer therapy and diagnosis are grouped according to their chemical characteristics (liposomal, inorganic, and polymeric). NPs execute multiple effects, including improved targeted delivery, and bioavailability, controlled release, reduced toxicity, and multifunctionality (e.g., for bioimaging). ‘Minus’ indicate decrease, ‘plus’ increase/improvement by NP-based application. Created with BioRender.com.

**Figure 5 ijms-25-05171-f005:**
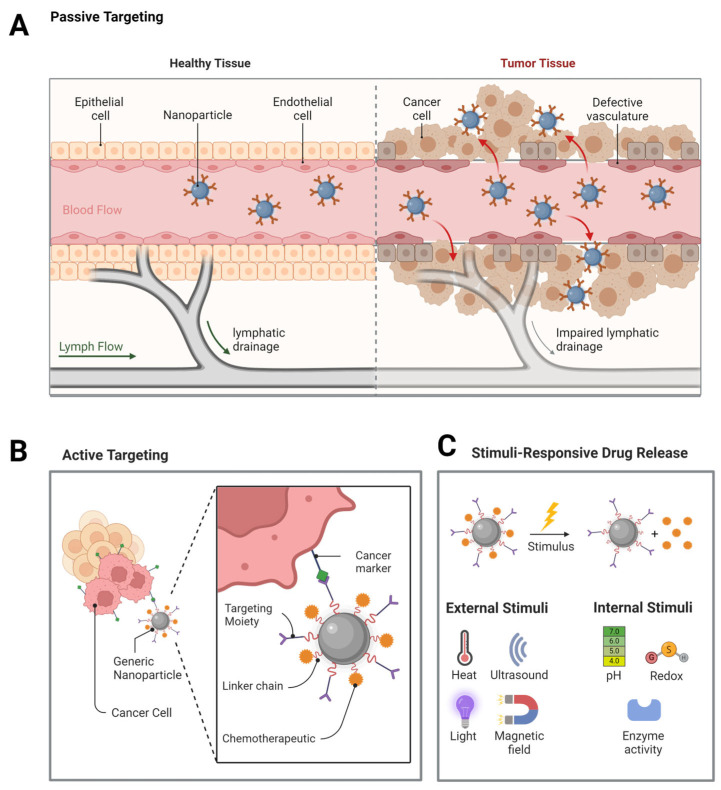
Targeting of cancer cells by nanotechnical delivery platforms. Examples for NPs used for DD are shown in (**A**). Loaded drugs can be delivered by EPR-based, passive targeting (**B**) or active targeting (**C**). Smart NPs allow for stimuli–response drug release, triggered by external or internal factors, such as pH or temperature. Created with BioRender.com.

**Figure 6 ijms-25-05171-f006:**
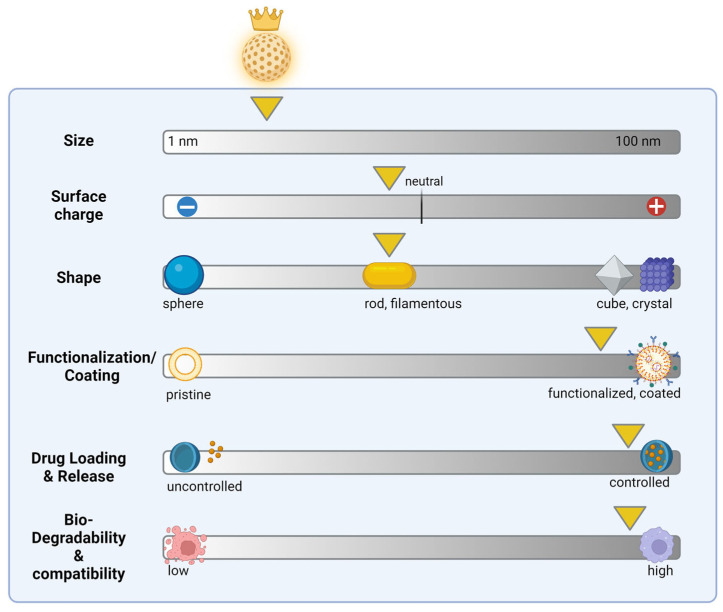
Suggested optimal physicochemical properties of nanoparticles used for cancer therapy. The physicochemical properties of NPs play a crucial role in their efficiency for therapeutic applications. Some of the key advantageous properties (marked by yellow arrow) include small size range, neutral or slightly negative surface charge, rod- or filamentous shape [152], targeted surface functionalization and coating (e.g., PEG), efficient encapsulation and controlled release of drug cargo, and high biodegradability and compatibility to minimize toxicity. Created with BioRender.com.

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
