# Peer review of "Revolution in Cancer Treatment: How Are Intelligently Designed Nanostructures Changing the Game?"

_ijms, 2024, doi:10.3390/ijms25105171_

Round 1

Reviewer 1 Report

Comments and Suggestions for Authors

This review aims to highlight the potency of nanoparticles for cancer treatment, emphasizing their design and application in overcoming biological barriers for effective therapy. The significance of nanoparticle design in DD by optimizing size, shape, and surface properties for better targeting was also discussed. Despite an attempt at covering the broad applications and properties of NP in cancer treatment, this review did not achieve a high level of detailed analysis in each section. One main issue of this article is a lack of focus on the basic working principles of various NPs in cancer therapy. The overall style of this review is broad, and sometimes vague, but such an issue may be addressed by narrowing down the topics that this article aims to cover. Therefore, I suggest reconsideration after major revision. 

Specific comments are as follows:

  1. Figures 1-3: These figures are very broad and have limited value due to a lack of comprehensive coverage in each respective topic. For example, Figure 2 shows different types of nanostructures and points them all to “Different Effects” without specifications. Figure 3 should include surface modifications that serve for other effects, such as biomimicry. In addition, check for spelling of “specific binding” in Figure 3. The diagram representing “Treatment” in Figure 1 should be more related to the application of NP.
  2. Section 4.1. Applications should highlight more on the recent advances, utilizing specific examples and studies. In line 418-420, common NP used in cancer treatments are mentioned and shown in Figure 5A without elaboration on what functions they serve. The NPs in Figure 5A should also be categorized by their function/purpose or characterization.
  3. There is general lack of discussion of molecular or cellular mechanisms through which NP impact cancer cells as well as the methodological approaches in nanoparticle research and their respective strengths and weaknesses. 
  4. Section 4: Instead of highlighting how underrepresented some NP-based cancer therapies are, devote more attention to providing a more comprehensive review on mechanisms and properties of NP in cancer treatment. 
  5. Section 3.1. Formation of Protein Corona should not be in the same category as the NP size, shape, and surface charge since it is not an intrinsic property of NP. 
  6. Some of important related articles are missing like Materials today 2020, 37, 112-125, etc

Comments on the Quality of English Language

Some format errors are found, and should be double-check carefully.

Author Response

Reviewer #1

This review aims to highlight the potency of nanoparticles for cancer treatment, emphasizing their design and application in overcoming biological barriers for effective therapy. The significance of nanoparticle design in DD by optimizing size, shape, and surface properties for better targeting was also discussed. Despite an attempt at covering the broad applications and properties of NP in cancer treatment, this review did not achieve a high level of detailed analysis in each section. One main issue of this article is a lack of focus on the basic working principles of various NPs in cancer therapy. The overall style of this review is broad, and sometimes vague, but such an issue may be addressed by narrowing down the topics that this article aims to cover. Therefore, I suggest reconsideration after major revision. 

Answer: We are pleased to learn that the referee considers our work as relevant for the field. We fully value his/her time and expertise, followed the reviewer’s suggestions, addressed all comments, and modified the revised manuscript accordingly (see detailed response below).

  1. Figures 1-3: These figures are very broad and have limited value due to a lack of comprehensive coverage in each respective topic. For example, Figure 2 shows different types of nanostructures and points them all to “Different Effects” without specifications. Figure 3 should include surface modifications that serve for other effects, such as biomimicry. In addition, check for spelling of “specific binding” in Figure 3. The diagram representing “Treatment” in Figure 1 should be more related to the application of NP.

We thank the reviewer for the suggestions, revised and re-ordered all Figures. Especially Figure 2 (now Figure 4) was specified, and Figure 3 (now Figure 2) was completely revised. Figure 1 should serve as an introductory overview summarizing possible applications for NPs in cancer therapy. We added Figure 6 to give a recapitulating overview of suggested optimal design of NPs for therapy.

Figure 1. Exploring nanoparticles (NPs) for cancer therapy. NPs are utilized for various purposes in cancer treatment, including tumor targeting, drug delivery, photo-/radio/ chemo-nanotherapy, where NPs act as active agents, not just as carriers, as well as imaging, and diagnosis. Created with BioRender.com.

Figure 2. Surface modifications of nanoparticles for cancer therapy. Surface modifications of NPs aiming at the improvement of biocompatibility, targeting ability, and therapeutic efficacy can be classified into four groups: (1) Biomimetic coatings are mainly made of different (bio)membranes mimicking biological identity; (2) Targeting Ligands: NPs are conjugated with targeting ligands such as antibodies, peptides, or aptamers to enhance specific binding to cancer cells or tumor vasculature; (3) Responsive coatings include different polymers which are sensitive towards acidic pH or enzyme present in the tumor microenvironment, or external heat resulting in controlled drug release; (4) Polyethylene Glycol (PEG)ylation of NPs improve their stability, reduce clearance by the immune system, and prolong circulation time in the bloodstream. Created with BioRender.com.

Figure 4. Advantages of nanoparticles (NPs) in cancer treatment. NPs commonly used for cancer therapy and diagnosis are grouped according to their chemical characteristics (liposomal, inorganic, and polymeric). NPs execute multiple effect, including improved targeted delivery, and bioavailability, controlled release, reduced toxicity, and multifunctionality (e.g. for bioimaging). Created with BioRender.com.

Figure 6. Suggested optimal physicochemical properties of nanoparticles used for cancer therapy. The physicochemical properties of NPs play a crucial role in their efficiency for therapeutic applications. Some of the key advantageous properties (marked by yellow arrow) include small size range, neutral or slightly negative surface charge, rod- or filamentous shape [155] targeted surface functionalization and coating (e.g. PEG), efficient encapsulation and controlled release of drug cargo, and high biodegradability and compatibility to minimize toxicity. Created with BioRender.com.

  1. Section 4.1. Applications should highlight more on the recent advances, utilizing specific examples and studies. In line 418-420, common NP used in cancer treatments are mentioned and shown in Figure 5A without elaboration on what functions they serve. The NPs in Figure 5A should also be categorized by their function/purpose or characterization.

We value this important remark which will increase comprehensibility of our review for a broad readership. We added specific functions of the mentioned NPs, and revised Figure 5 (categorized NPs now shown in Figure 4). Furthermore, we believe that our review covers relevant recent advances of the field, e.g. smart NPs, TME-responsive nanostructures, as well as extra-cellular vesicles (exosomes).

…Key NPs for cancer applications include nanoliposomes, nanoemulsions, polymeric NPs, quantum dots, gold, iron and carbon NPs [98]. In addition to general advantageous properties, each NP class possesses specific features that can be utilized for applications in nanomedicine (Figure 4). Liposomes and emulsions show ideal membrane delivery properties resulting in decreased drug toxicity and increased efficacy [99]. Whereas polymeric NP are used for their prolonged circulation time [100], quantum dots are versatile bioimaging tools, especially for cancer imaging[101]. The application of inorganic NPs, such as gold, iron and carbon NPs are manifold, including all fields of nanomedicine. The advantageous features of gold NPs encompass high chemical stability, and thus biocompatibility, as well as high tunability. In addition to the previously mentioned features, iron NPs also exhibit unique magnetic properties, which make them particularly suitable for use as contrast agents in imaging procedures [102]. Due to their high degree of biodegradability and biocompatibility, as well as their diversity, peptide-based NPs offer versatile tools for biomedical applications, such as tissue engineering, regenerative medicine, bioimaging, antibacterial applications, and of course cancer treatment [103]…

Figure 5. Targeting of cancer cells by nanotechnical delivery platforms. Examples for NPs used for DD are shown in (A). Loaded drugs can be delivered by EPR-based, passive targeting (B) or active targeting (C). Smart NPs allow for stimuli-response drug release, triggered by external or internal factors such as pH, or temperature (D). Created with BioRender.com.

  1. There is general lack of discussion of molecular or cellular mechanisms through which NP impact cancer cells as well as the methodological approaches in nanoparticle research and their respective strengths and weaknesses. 
  2. Section 4: Instead of highlighting how underrepresented some NP-based cancer therapies are, devote more attention to providing a more comprehensive review on mechanisms and properties of NP in cancer treatment. 

We thank the referee for this comment highlighting the importance of general molecular mechanisms of nanoformulated drugs. Some important mechanisms are mentioned in the revised manuscript, e.g.

Apoptosis induction by ROS generation: …Nevertheless, if the intracellular concentration of Zn2+ surpasses the cell's tolerance threshold, it can harm intracellular mitochondria and trigger an outburst of reactive oxygen species (ROS), resulting in severe intracellular oxidative stress known as "Zn2+ interference"…

Effect on intracellular signalling pathways: …Effective delivery of Oxa and miR-320 to the nucleus and cytoplasm, respectively, was achieved by these NPs. This study is the initial one to showcase the simultaneous intracellular adjustment of the NRP1/Rac1, PI3K/Akt/mTOR, GSK-3β/FOXM1/β-catenin, P-gp/MRPs, KRAS/Erk/Oct4/Yap1, and N-cadherin/Vimentin/Slug pathways in order to impede the proliferation, advancement, and resistance to multiple drugs in cancerous cells.  …

Increase of treatment effectiveness: .. The NPs facilitated the uptake of radiosensitizers into cancer cells, leading to cell arrest and synergistic enhancement of radiation-induced DNA damage and apoptosis….

Hypoxia: .. Chen et al. engineered a nano-biomimetic DD vehicle which does not only target bone and cancer tissue, but also shows immune escape ability [124]. Here, tumor toxicity can be triggered by hypoxia at the tumor site during a photothermal treatment process [124]…

Immunomodulation of the TME: Zhou et al have addressed this shift of HNC therapy towards immunotherapy in a recent study [126]. The authors designed an extracellular vesicle-based intelligent nano-platform, which is able to respond to and modulate the tumor microenvironment, thereby inducing immunogenic cell death…

Additionally, we refer to other works on that topic.

In the following section, we would like to give an introduction of general strategies of nanoparticulate drug systems for cancer therapy. By focusing on innovative approaches of recent studies, we want to give new impetus especially for the treatment of head and neck tumors. For a more general analysis of NP-based drugs and their molecular mechanisms, please refer to other reviews on that topic [94–96].

  1. Section 3.1. Formation of Protein Corona should not be in the same category as the NP size, shape, and surface charge since it is not an intrinsic property of NP. 

We agree with this important comment, and apologize for this misunderstanding. To clarify our story, we completely revised and re-ordered section 3 which is now focusing on the relevance of the protein corona and influencing NPs characteristics.

3.3. charge and chemistry

  1. Some of important related articles are missing like Materials today 2020, 37, 112-125, etc

We value the referee's opinions, and included several relevant articles.

By focusing on innovative approaches of recent studies, we want to give new impetus especially for the treatment of head and neck tumors. For a more general analysis of NP-based drugs and their molecular mechanisms, please refer to other reviews on that topic [94–96].

In summary, we are highly thankful for the constructive review of our manuscript, which helped us to improve the quality of the revised manuscript. By modifying the manuscript accordingly, we hope that the revised manuscript is now considered acceptable by reviewer #1 for publication in IJMS.

Reviewer 2 Report

Comments and Suggestions for Authors

In this manuscript, the authors provide an overview of the properties and application of nanoparticles (NPs) for cancer therapy. I have some comments for the authors for revising this manuscript. Overall, the toxicity and limitations of NPs have not been critically discussed in this manuscript.

1. Are there any FDA-approved NPs for cancer treatments? If yes, the authors should provide some discussions and/or tables for these FDA-approved NPs.

2. While many benefits of NPs are discussed, details on the toxicity and other limitations of NPs are lacking. Please provide some current methods to reduce the toxicity of these NPs and future directions for research.

3. The authors could provide more detailed information on what is believed to be the optimal physicochemical properties of NPs for efficient therapy. Example size, surface charge, shape, etc. And collect all the information in a diagram that is immediately readable.

4. Some suitable reading materials and recent references for authors to improve the manuscript could be:

a) Zhou, Z. H., Zhou, X. Y., Zhang, Y. Y., Zhao, T. C., Li, J., Zhong, L. P., Pang, Y. C. (2024). Macrophage‐Capturing Self‐Assembly Photosensitizer Nanoparticles Induces Immune Microenvironment Re‐Programming and Golgi‐Responsive Immunogenic Cell Death in Head AND Neck Carcinoma. Advanced Healthcare Materials, 2400012.

b) Taratula, O., Korzun, T., Jozic, A., Grigoriev, V., Sahay, G., Marks, D. L. (2024). Abstract LB019: Comprehensive therapeutic impact: Follistatin mRNA-loaded lipid nanoparticles halt lung metastases and prolong survival in murine head & neck squamous cell carcinoma. Cancer Research, 84(7_Supplement), LB019-LB019.

c) Persano, F., Nobile, C., Piccirillo, C., Gigli, G., Leporatti, S. (2022). Monodisperse and nanometric-sized calcium carbonate particles synthesis optimization. Nanomaterials, 12(9), 1494.

d) Gavas, S., Quazi, S., Karpiński, T. M. (2021). Nanoparticles for cancer therapy: current progress and challenges. Nanoscale research letters, 16(1), 173.

e) Persano, F., Leporatti, S. (2020). Current Overview of inorganic nanoparticles for the treatment of central nervous system (CNS) diseases. Current Nanomaterials, 5(2), 92-110.

f) Vanbilloen, W. J., Rechberger, J. S., Anderson, J. B., Nonnenbroich, L. F., Zhang, L., Daniels, D. J. (2023). Nanoparticle Strategies to Improve the Delivery of Anticancer Drugs across the Blood–Brain Barrier to Treat Brain Tumors. Pharmaceutics, 15(7), 1804.

g) Chen, H., Ji, J., Zhang, L., Luo, C., Chen, T., Zhang, Y., et al. (2024). Nanoparticles Coated with Brain Microvascular Endothelial Cell Membranes can Target and Cross the Blood–Brain Barrier to Deliver Drugs to Brain Tumors. Small, 2306714.

Author Response

Reviewer #2

In this manuscript, the authors provide an overview of the properties and application of nanoparticles (NPs) for cancer therapy. I have some comments for the authors for revising this manuscript. Overall, the toxicity and limitations of NPs have not been critically discussed in this manuscript.

Answer: We are pleased to learn that the referee considers our work as relevant for the field. We fully value his/her time and expertise, followed the reviewer’s suggestions, addressed all comments, and modified the revised manuscript accordingly (see detailed response below).

  1. Are there any FDA-approved NPs for cancer treatments? If yes, the authors should provide some discussions and/or tables for these FDA-approved NPs.

Important question! As prominent example, we mentioned Abraxane as FDA-approved NP-drug. In the revised manuscript we added other examples and also references for detailed work on that.

…One of the biggest success stories of NPs in cancer therapy up until now is Abraxane. Abraxane contains nano-formulated paclitaxel in a human serum albumin (HSA) shell and was introduced as the first ever NP drug in 2005 after approval by the FDA in the US. Its indications include for example breast cancer, lung cancer, and pancreatic cancers [101]. It has also been reported in the literature that NP albumin-bound paclitaxel pro-vides more effective paclitaxel delivery to tumor cells with fewer side effects than tradi-tional chemotherapy strategies [102]. Clinical studies showed evidence for reduced sys-tematic toxicity of paclitaxel nanoformulations, but often lack in proper study design [95].

Besides Abraxane, there are also other nanoformulated drugs with various characteristics approved by the FDA. Within the group of metallic NPs, Feraheme, a drug containing iron oxide NP formulation is approved for the treatment of metastatic prostate and testicular cancer [94].Further examples are Doxil (liposomal doxorubicin), Eligard (poly (DL-lactide-coglycolide)), DaunoXome (liposomal daunorubicin), or Onivyde (liposomal irinotecan) which are approved for the treatment of different solid cancers or leukemias (for details see [94])….

  1. While many benefits of NPs are discussed, details on the toxicity and other limitations of NPs are lacking. Please provide some current methods to reduce the toxicity of these NPs and future directions for research.

We value the referee´s remark, and completely agree that toxicity and limitations of NPs have to be discussed. We summarized important limitations in the ‘Conclusion’ section. Methods to reduce toxicity and increase biocompatibility are also included in revised Figure 2 (previous Figure 3).

Nevertheless, numerous obstacles remain in enhancing the clinical application of NMs. Important aspects of limitations in NP-based therapies include (1) their biocompatibility and toxicity: Some NPs may cause immune reactions or toxicity in the body, limiting their clinical application. The long-term effects of NPs on human health are still not fully understood. (2) Off-target effects: Despite their targeted delivery capabilities, NPs may still accumulate in healthy tissues, leading to off-target effects and potential toxicity. (3) Biodegradability: Many NPs have limited biodegradability, which may result in their accumulation in the body over time, potentially causing long-term adverse effects.(4) Cost and scalability: The production of NPs for clinical use can be expensive, and large-scale production may be challenging. This can limit their widespread use in cancer treatment. (5) Complexity of design and characterization: Designing NPs with the desired properties for targeted drug delivery can be complex, and characterizing their structure and function is often challenging. (6) Drug loading and release efficiency: Ensuring efficient loading of drugs onto nanoparticles and controlling their release kinetics can be difficult, affecting the efficacy of the treatment. (7) Resistance: Cancer cells may also develop resistance to NP-based therapies over time, reducing their effectiveness.

Addressing these limitations requires further research and development to optimize the design, synthesis, and application of NPs in cancer treatment [153].

Figure 2. Surface modifications of nanoparticles for cancer therapy. Surface modifications of NPs aiming at the improvement of biocompatibility, targeting ability, and therapeutic efficacy can be classified into four groups: (1) Biomimetic coatings are mainly made of different (bio)membranes mimicking biological identity; (2) Targeting Ligands: NPs are conjugated with targeting ligands such as antibodies, peptides, or aptamers to enhance specific binding to cancer cells or tumor vasculature; (3) Responsive coatings include different polymers which are sensitive towards acidic pH or enzyme present in the tumor microenvironment, or external heat resulting in controlled drug release; (4) Polyethylene Glycol (PEG)ylation of NPs improve their stability, reduce clearance by the immune system, and prolong circulation time in the bloodstream. Created with BioRender.com.

  1. The authors could provide more detailed information on what is believed to be the optimal physicochemical properties of NPs for efficient therapy. Example size, surface charge, shape, etc. And collect all the information in a diagram that is immediately readable.

We are thankful for this great suggestion, and added a section including new Figure 6 in the conclusion section.

Conversely, these findings also allow for conclusions to be drawn regarding the optimal properties of NPs. Optimal physicochemical properties may vary depending on the specific therapeutic application and the targeted tissue or cell type, but NPs with a small size [2], neutral or slightly negative surface charge [154], appropriate surface functionalization, shape, surface coating, drug loading and release capabilities, as well as biodegradability and biocompatibility are believed to be optimal for efficient therapy (Figure 6).

Figure 6. Suggested optimal physicochemical properties of nanoparticles used for cancer therapy. The physicochemical properties of NPs play a crucial role in their efficiency for therapeutic applications. Some of the key advantageous properties (marked by yellow arrow) include small size range, neutral or slightly negative surface charge, rod- or filamentous shape [155]targeted surface functionalization and coating (e.g. PEG), efficient encapsulation and controlled release of drug cargo, and high biodegradability and compatibility to minimize toxicity. Created with BioRender.com.

  1. Some suitable reading materials and recent references for authors to improve the manuscript could be:
  2. a) Zhou, Z. H……

We thank the reviewer for this comprehensive list of additional references, added relevant work, and completely revised our bibliography.

Especially for the challenging treatment of brain tumors, researchers designed advanced NP-based systems allowing passage of the BBB and thus delivery of drugs to tumor cells of the brain [97].

Besides the established treatment of HNC based on (radio)chemotherapy, combined treatment schedules including immune checkpoint inhibitors, such as pembrolizumab have revealed improved prognosis in clinical studies, offering new hope to patients with advanced or recurrent disease [125]. Zhou et al have addressed this shift of HNC therapy towards immunotherapy in a recent study [126]. The authors designed an extracellular vesicle-based intelligent nano-platform, which is able to respond to and modulate the tumor microenvironment, thereby inducing immunogenic cell death. Here, especially the use of extracellular vesicles that mimic natural occurring exosomes offer significant and distinctive advantages over synthetic NPs, such as good cell delivery properties and reduced immune response…

In summary, we are highly thankful for the constructive review of our manuscript, which helped us to improve the quality of the revised manuscript. By modifying the manuscript accordingly, we hope that the revised manuscript is now considered acceptable by reviewer #2 for publication in IJMS.

Round 2

Reviewer 1 Report

Comments and Suggestions for Authors

No comment anymore

Reviewer 2 Report

Comments and Suggestions for Authors

The authors have significantly improved the manuscript that can be accepted for publication in the present form.